# EFFICIENT AND INTERPRETABLE MULTI-AGENT LLM ROUTING VIA ANT COLONY OPTIMIZATION

## ABSTRACT

The instruction-following and semantic understanding capabilities of large language models (LLMs) serve as the core competence of Multi-Agent Systems (MAS), enabling collective strategy coordination. However, task routing in MAS remains a critical performance bottleneck, especially in dynamic and resource-constrained environments. Existing LLM-based routing approaches often suffer from limited transparency, static allocation strategies, and insufficient system state awareness. To address these challenges, we propose AMRO: Ant colony inspired Multi-agent Routing Optimization. AMRO models agent interactions as a function-based directed graph, utilizing a pheromone-driven node update mechanism and an adaptive pheromone decay strategy to achieve real-time perception and response to environmental changes, thereby continuously optimizing routing assignments. This approach significantly enhances routing efficiency and overall system performance, while the pheromone-guided path selection offers strong interpretability for the routing process. We conduct extensive experiments on five public benchmark datasets. The results show that AMRO achieves an average improvement of 1.97% in pass@1 accuracy over the baseline and demonstrates superior efficiency and robustness under high concurrency. These findings indicate that AMRO provides an efficient and interpretable solution to the routing problem in LLM-based MAS.

## 1 INTRODUCTION

MAS is a distributed system composed of multiple LLM-based agents. With LLMs as their core capability carriers, these agents possess abilities such as natural language understanding and generation, contextual reasoning, knowledge representation, and decision-making planning. Through mutual communication, collaboration, competition, or coordination, they collectively accomplish complex tasks or achieve system goals (Alonso et al., 2001; Li et al., 2025). MAS can decompose complex tasks and delegate them to specialized agents, showing strong performance and scalability in domains such as automated programming (Yang et al., 2025; Yuan et al., 2024), mathematical reasoning (Motwani et al., 2024), and collaborative decision-making (Jin et al., 2025; Wang et al., 2024c). However, in dynamic and resource-constrained environments, task routing becomes a critical performance bottleneck. The core objective of task routing is to select the most appropriate agent to fulfill a request, based on task semantics and system state.

Current MAS routing methods typically rely on LLMs to understand and match task semantics, routing requests to the most suitable agents (Chuang et al., 2025; Wang et al., 2024b). Some approaches dynamically select model nodes with different capabilities to balance computational cost and response quality, while others explore multi-stage collaboration mechanisms to optimize the distribution process of complex tasks (Wang et al., 2024a). Despite these advancements, existing methods exhibit notable limitations. First, many strategies depend heavily on the implicit reasoning of black-box LLMs, lacking transparent explanations for routing decisions—problematic in high-stakes domains such as healthcare and finance (Marey et al., 2024; Ramlochan, 2023). Second, most existing methods adopt static or semistatic allocation strategies, which struggle to adapt in real time to changes in node load, network fluctuations, or task dynamics, leading to unstable performance in complex settings. Additionally, the dependence on large-scale annotated data and costly training procedures poses major deployment challenges in edge computing or low-latency scenarios (Varangot-Reille et al., 2025; Zhang et al., 2024b). Although reward-based and meta-learning

strategies have been explored to improve adaptability and efficiency (Hu et al., 2024a; 2023), these solutions often involve complex designs and high training costs, hindering widespread adoption.

To address these issues, we propose AMRO, a routing optimization approach based on the Ant Colony Optimization (ACO) algorithm (Dorigo et al., 2007). AMRO models MAS as a directed graph, where nodes represent agents with distinct functions, and edges denote connection relationships between agents, reflecting possible paths for task flow or information transmission. During the routing planning process, AMRO designs a pheromone-guided probabilistic selection strategy that integrates node capabilities, load, response speed, and resource consumption to guide subsequent node selection. AMRO employs a global pheromone decay mechanism, uniformly decaying the pheromone on all edges proportionally at the end of each epoch. This strategy aims to suppress infinite pheromone accumulation, prevent the search from falling into local optima, and enhance the ability to explore new high-quality paths. Additionally, by combining with a fitness function, it achieves a dynamic balance between positive feedback and global search. AMRO can dynamically respond to environmental changes and converge to optimal task allocation schemes in real time.

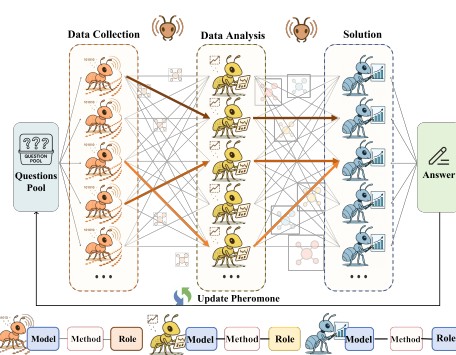

Figure 1: Tasks are routed through three stages: collection, analysis, and solution, via probabilistic paths guided by pheromone signals. After execution, high-quality paths receive reinforced pheromones, increasing their selection likelihood in future iterations.

Our main contributions are summarized as follows:

- We propose the first integration of ACO into routing for LLM-based MAS.

- By developing a pheromone-based feedback mechanism while designing a load-balancing routing selection mechanism, we enable the system to achieve high-performance and efficient routing planning.

- Addressing the limitations of the black-box selection process in existing routing planning, we leverage the visualizable characteristics of pheromones to propose a novel transparency solution, significantly enhancing the system's interpretability. Concurrency tests and visualization experiments confirm AMRO's interpretability as well as stability.

- Experimental results on five datasets demonstrate that AMRO exhibits competitive performance against existing methods.

## 2 RELATED WORK

### 2.1 LLM-BASED MULTI-AGENT SYSTEM ROUTING

MAS are composed of multiple agents with autonomous perception, learning, and decision-making capabilities, completing complex tasks through distributed collaboration (Dorri et al., 2018). They overcome the limitations of single agent systems in memory capacity and scalability (Balaji & Srinivasan, 2010). LLM-based MAS integrate the powerful language understanding capabilities of LLMs (Kasneci et al., 2023; Zheng et al., 2025) and group strategy coordination abilities (Li et al., 2024; Han et al., 2024), further enhancing their problem-solving capacity for complex tasks. To improve system efficiency, LLM routing precisely allocates user requests to appropriate subagents, tools, plugins, or modules based on task content (Hu et al., 2024b), making the design of effective routing strategies a current research focus. AGENTVERSE (Chen et al.) dynamically determines the composition of the agent through an expert recruitment phase. MAD (Liang et al., 2023) designs a multi-agent debate structure with sparse communication topologies, achieving comparable performance while significantly reducing computational costs. However, non-learnable path strategies in complex tasks restrict model generalization and flexibility. ZOOTER (Lu et al., 2023) proposes reward-guided routing, extracting rewards from training queries to train a routing function that as-

signs each query to an LLM with relevant expertise. RouterDC (Chen et al., 2024) learns a query-based router using sample-LLM and inter-sample contrastive loss functions. Hybrid-LLM (Ding et al., 2024) introduces a hybrid LLM routing method to improve reasoning efficiency by combining the advantages of multiple LLMs. RouteLLM (Ong et al., 2024) optimizes the balance between cost and response quality through dynamic selection of strong and weak LLMs, while MasRouter (Yue et al., 2025) addresses complex routing problems using a three-level cascaded framework for collaboration mode determination, role allocation, and routing assignment.

Despite the good performance achieved by the above methods, practical applications still demand higher training efficiency and accuracy. Additionally, the black-box nature of LLMs limits the interpretability of routing. To address these issues, we introduce ACO and design a multi-agent routing mechanism, enabling the MAS to maintain low cost, high efficiency, and high concurrent processing capabilities while enhancing interpretability.

### 2.2 HEURISTIC PATH OPTIMIZATION

Heuristic path optimization rapidly searches for optimal or near-optimal paths through empirical strategies (Tan et al., 2021; Yahia & Mohammed, 2023). Classic heuristic path optimization algorithms include genetic optimization algorithms (Sivanandam et al., 2008), simulated annealing algorithms (Rutenbar, 1989), and particle swarm optimization algorithms (Wang et al., 2018; Gad, 2022), among others. The ant colony algorithm, in particular, provides effective optimization strategies for fields such as path planning (Cui et al., 2024), network routing, and vehicle scheduling due to its feedback mechanism and strong parallel computing characteristics. ACO algorithm is an optimization method inspired by the foraging behavior of natural ant colonies (Blum, 2005; Dorigo & Socha, 2018). In nature, ants indirectly communicate by releasing pheromones while searching for food. Other ants prefer paths with higher pheromone concentrations, as these typically indicate better routes. This mechanism forms a positive feedback loop, guiding more ants to follow optimal paths until the colony identifies the shortest route from the nest to food sources. AddACO (Scianna, 2024) proposed incorporating decision rules based on linear convex combinations into the ant colony algorithm, improving the computational efficiency for the Traveling Salesman Problem (TSP). DYACO (Liang et al., 2024) optimized the impact of complex slopes in deep-sea mining areas on path planning by dynamically adjusting key information such as heuristic guidance, significantly enhancing the convergence speed of path optimization. PACO (Si & Bao, 2024) addressed the local optimum problem of traditional ACO through improved pheromone update methods and hybrid strategies, and substantially boosted path planning efficiency via parallel computing.

## 3 METHOD

This paper introduces ant colony optimization into LLM-based MAS routing planning for the first time. By quantifying node parameters, designing a pheromone-guided node selection method, and introducing a pheromone update and decay mechanism, we construct a multi-parallel routing planning system for LLM-based MAS in complex scenarios. This provides a novel bio-inspired solution for traffic allocation and node selection in LLM-based MAS routing planning.

Ant Colony Optimization (ACO) is a representative swarm intelligence optimization algorithm, whose core lies in simulating the cooperative search process of ants via pheromone communication. The algorithm mainly comprises three mechanisms, probabilistic path selection, pheromone update and evaporation, and positive feedback and diversity maintenance.

### 3.1 PRELIMINARIES

**Probabilistic Path Selection.**

Each "ant" incrementally selects nodes in the solution space, with the probability of choosing a path determined by both pheromone concentration and heuristic factors (such as distance or cost). Pheromones reflect historical experience, while heuristics capture local optimality, guiding ants to balance global and local search.

**Pheromone Update and Evaporation.** After each iteration, ants reinforce the pheromone levels on the paths according to the quality of the constructed solutions (positive feedback). Meanwhile,

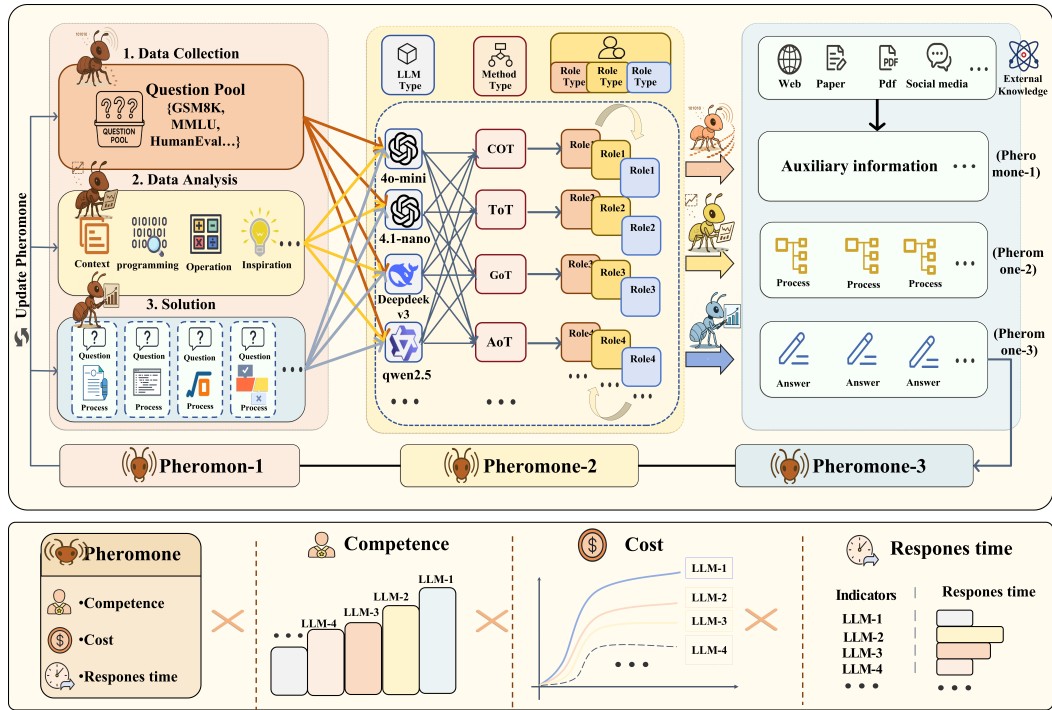

Figure 2: Three-layer task routing in the AMRO architecture. AMRO employs a three-layer routing strategy using model-method-role combinations. The first layer gathers external knowledge, the second analyzes the task, and the third synthesizes the answer. Routing is guided by pheromone-based probabilities and refined through multi-level feedback for dynamic agent coordination.

pheromones on all paths evaporate at a certain rate, preventing excessive accumulation and premature convergence.

**Positive Feedback and Diversity Maintenance.** High-quality paths accumulate more pheromones due to frequent selection, forming a positive feedback loop that accelerates the discovery of optimal solutions. Pheromone evaporation and probabilistic selection help maintain solution diversity and enhance global search capability. Through these mechanisms, ACO enables distributed, adaptive, and efficient search in complex combinatorial optimization and path planning problems, exhibiting strong robustness and scalability

### 3.2 AMRO ALGORITHM

In this study, we propose an AMRO (see Figure 1 and 2), algorithm-based approach for routing in MAS. We model the routing problem as a layered directed graph and apply AMRO to optimize the paths for task transmission between agents. The methodology includes problem modeling, path selection, pheromone update, and load management.

#### 3.2.1 PROBLEM MODELING

**Graph Structure and Path Definition.** We model the MAS as a hierarchical directed graph $G = (V, E)$. Specifically, the graph consists of $N$ layers, each containing $n$ nodes. The node set at the $l$-th layer is defined as: $V_l = \{v_{l,1}, v_{l,2}, \ldots, v_{l,n}\}$, so the total node set is $V = \bigcup_{l=1}^{N} V_l$. The directed edge set $E$ contains only the pairs of nodes between adjacent layers: $E = \{(v_{l,i}, v_{l+1,j}) \mid 1 \leq l < N, 1 \leq i \leq n, 1 \leq j \leq n\}$, where $(v_{l,i}, v_{l+1,j})$ represents the directed edge from the $i$-th node in the $l$-th layer to the $j$-th node in the $l + 1$-th layer. Each node $v_{l,j}$ corresponds to a functional agent, such as a question-answering system, search, or inference engine.

Each edge $(v_{l,i}, v_{l+1,j})$ represents the ability to route tasks from an agent in the $l$-th layer to an agent in the $l+1$-th layer.

**Task Description.** The task is a single-objective task, where it must route from a source node $v_{1,s}$ (a node in the first layer) to a target node $v_{N,d}$ (a node in the final layer), and the routing can only move along directed edges between adjacent layers. The path is defined as $P = [v_{1,s}, v_{2,k_2}, v_{3,k_3}, \ldots, v_{N,d}]$, where $v_{1,s}$ denotes the source node in the first layer (start), with $s \in \{1, 2, \ldots, n\}$. For intermediate layers, $v_{l,k_l}$ denotes the $k_l$-th node in the $l$-th layer, where $2 \leq l \leq N-1$ and $k_l \in \{1, 2, \ldots, n\}$. Finally, $v_{N,d}$ denotes the target node in the $N$-th layer (end), with $d \in \{1, 2, \ldots, n\}$. The $P$ represents a task starting from the source node in the first layer, passing through one node in each layer, and finally reaching the target node in the last layer. For each pair of adjacent nodes $(v_{l,k_l}, v_{l+1,k_{l+1}})$ along the path, there is a directed edge, and the routing strictly follows the layer-wise order (i.e., each step moves from layer $l$ to layer $l+1$).

### 3.2.2 PHEROMONE AND HEURISTIC FACTOR DEFINITIONS

**Pheromone Intensity.** The pheromone intensity $\tau_{l,i,j}$ is defined as the amount of pheromone on the directed edge $(v_{l,i}, v_{l+1,j})$ from the $i$-th node in layer $l$ to the $j$-th node in layer $l+1$.

**Heuristic Information.** The heuristic information $\eta_{l,i,j}$ is defined as the inverse of the response time from the source node $v_{l,i}$ to the target node $v_{l+1,j}$ for the directed edge $(v_{l,i}, v_{l+1,j})$. Specifically, it is defined as:

$$\eta_{l,i,j} = \frac{1}{\text{ResponseTime}(v_{l,i}, v_{l+1,j}) + \epsilon}, \tag{1}$$

where $\text{ResponseTime}(v_{l,i}, v_{l+1,j})$ is the average response time of edge $(v_{l,i}, v_{l+1,j})$, and $\epsilon$ is a small constant to prevent division by zero.

**Edge Cost and Normalized Unit Edge Cost.** For each edge $(v_{l,i}, v_{l+1,j})$, let $\text{UnitCost}(v_{l,i}, v_{l+1,j})$ denote its token consumption. The total outgoing cost at layer $l$ is

$$S_l = \sum_{i,j} \text{UnitCost}(v_{l,i}, v_{l+1,j}). \tag{2}$$

and the normalized unit edge cost is defined as:

$$\lambda_{l,i,j} = \frac{\text{UnitCost}(v_{l,i}, v_{l+1,j})}{S_l}. \tag{3}$$

**System Load and Decay.** The normalized current load is defined as:

$$\alpha_{l,i,j} = \frac{\text{Tasks}(v_{l+1,j})}{\text{Capacity}(v_{l+1,j})}, \tag{4}$$

where $\text{Tasks}(v_{l+1,j})$ is the number of tasks assigned to node $v_{l+1,j}$, and $\text{Capacity}(v_{l+1,j})$ is the maximum task processing capacity of node $v_{l+1,j}$. The system load decay factor is defined as: $\gamma_{l,i,j} = e^{-\beta \alpha_{l,i,j}}$, where $\beta > 0$ is a parameter controlling the decay rate.

**Node Capability and Edge Weight.** The capability of the model at node $v_{l+1,j}$ is defined as $\text{Ability}(\text{Model}_{l+1,j})$, and the total ability of all models is given by:

$$W = \sum_{k=1}^{N} \text{Ability}(\text{Model}_k). \tag{5}$$

The weight of the edge from $v_{l,i}$ to $v_{l+1,j}$ is defined as:

$$s_{l,i,j} = \frac{\text{Ability}(\text{Model}_{l+1,j})}{W}. \tag{6}$$

**Edge Selection Probability.** In the hierarchical directed graph, the probability $p_{l,i,j}$ that an ant moves from the $i$-th node $v_{l,i}$ in the $l$-th layer to the $j$-th node $v_{l+1,j}$ in the $l+1$-th layer is defined as:

$$p_{l,i,j} = \frac{\left[\tau_{l,i,j}(t)\right]^{\alpha} \left[\eta_{l,i,j}\right]^{\beta} \left[\gamma_{l,i,j}\right]^{\delta} \left[s_{l,i,j}\right]^{\lambda} \left[l_{l,i,j}\right]^{-\mu}}{\sum_{k=1}^{n} \left[\tau_{l,i,k}(t)\right]^{\alpha} \left[\eta_{l,i,k}\right]^{\beta} \left[\gamma_{l,i,k}\right]^{\delta} \left[s_{l,i,k}\right]^{\lambda} \left[l_{l,i,k}\right]^{-\mu}}. \tag{7}$$

where $\alpha, \beta, \delta, \lambda, \mu$ are the weight parameters, and $\mu > 0$ represents the negative exponent of cost.

**Path Fitness Function.** To evaluate the quality of the path $P_k$ found by the ant, the following fitness function is introduced:

$$f(P_k) = w_1 \cdot \text{Delay}(P_k) + w_2 \cdot \big(1 - \text{Success}(P_k)\big)$$
$$+ w_3 \cdot \text{Load}(P_k). \tag{8}$$

We defined Total Token Consumption as $\text{Delay}(P_k) = \sum_{(l,i,j) \in P_k} l_{l,i,j}$, Cumulative Task Success Rate as $\text{Success}(P_k) = \prod_{(l,i,j) \in P_k} s_{l,i,j}$, Cumulative Task Success Rate as $\text{Success}(P_k) = \prod_{(l,i,j) \in P_k} s_{l,i,j}$ and Average Load as $\text{Load}(P_k) = \frac{1}{|P_k|} \sum_{(l,i,j) \in P_k} \gamma_{l,i,j}$. Where $|P_k|$ is the length of the path, and $w_1, w_2, w_3$ are the weight parameters to balance the importance of token consumption, success rate, and load in the fitness function.

**Pheromone Update Strategy.** After each iteration, the pheromone increment $\Delta\tau_{l,i,j}^k$ on the edge $(v_{l,i}, v_{l+1,j})$ is inversely proportional to the path fitness, specifically defined as:

$$\Delta\tau_{l,i,j}^k = \begin{cases} \dfrac{Q}{f(P_k) + \epsilon}, & \text{if } (v_{l,i}, v_{l+1,j}) \in P_k \\ 0, & \text{otherwise} \end{cases} \tag{9}$$

The global pheromone update rule is defined as:

$$\tau_{l,i,j}(t+1) = (1 - \rho) \cdot \tau_{l,i,j}(t) + \sum_{k=1}^{m} \Delta\tau_{l,i,j}^k, \tag{10}$$

where $\rho$ is the pheromone evaporation rate, and $m$ is the total number of ants in the current iteration. This strategy reduces outdated pheromone traces while reinforcing high-quality paths (see Appendix A.3 for pseudocode).

## 4 EXPERIMENT

### 4.1 EXPERIMENTAL SETUP

**Dataset and Benchmarks.** We validated the model on five public datasets, including GSM8K (Cobbe et al., 2021), MMLU (Hendrycks et al., 2020), MATH (Hendrycks et al., 2024), HumanEval (Chen et al., 2021), and MBPP (Austin et al., 2021). GSM8K is a dataset of 8.5K high-quality, linguistically diverse primary school math word problems, while MMLU covers 57 distinct categories ranging from basic knowledge to advanced professional disciplines. MATH, a math competition problem dataset, provides complete step-by-step solutions for each problem to train models to generate answer derivation processes and explanations. HumanEval is designed for evaluating code-generation models, containing programming problems with function signatures, docstrings, function bodies, and multiple unit tests, whereas MBPP consists of short Python programs crowdsourced from individuals with basic Python knowledge. These datasets collectively enable comprehensive assessment of the model's performance across mathematical reasoning, domain-specific knowledge, code generation, and problem-solving capabilities.

**Implementation Details.** We adopted the widely used pass@K metric in the industry as the evaluation standard for model performance, where the pass@1 metric is defined as whether the model can successfully output the correct answer or a code solution meeting the requirements on its first generation. This metric concisely and intuitively reflects the model's immediate effectiveness and practical application value in no-trial scenarios. To comprehensively compare model performance, the experiments covered single-agent methods (including CoT (Wei et al., 2022) (Chain of Thought), ComplexCoT (Fu et al., 2022), SC(COT) (Wang et al., 2022), SC(ComplexCoT)) (Wang et al., 2022), multi-agent methods without routing support (encompassing LLM-debate, GPTSwarm (Zhuge et al., 2024), Agent-prune (Zhang et al., 2024a), Chain (Qian et al., 2024), Tree (Yao et al., 2023), Complete Graph (Besta et al., 2024), AFlow (Zhang et al.)), single-agent routing methods (including FrugalGPT (Chen et al., 2023), PromptLLM (Feng et al., 2024), RouteLLM (Ong et al., 2024), RouterDC (Chen et al., 2024)), and multi-agent routing methods represented by MASRouter (Yue et al., 2025).

Table 1: Reordered performance of models using `gpt-4o-mini`, *Mul.* represents MAS and *Rout.* represents utilizing the routing.

| Method | Mul. | Rout. | GSM8K | MATH | MMLU | HumanEval | MBPP | Avg. |
|---|---|---|---|---|---|---|---|---|
| GPT-4o-mini | | | | | | | | |
| Vanilla | N | N | 93.17% | 66.09% | 77.81% | 85.71% | 72.20% | 79.00% |
| CoT | N | N | 93.68% | 67.24% | 78.43% | 86.69% | 69.60% | 79.13% |
| ComplexCoT | N | N | 93.43% | 67.05% | 81.05% | 87.58% | 75.80% | 80.98% |
| SC(CoT) | N | N | 93.32% | 66.28% | 81.05% | 87.58% | 73.00% | 80.25% |
| SC(ComplexCoT) | N | N | 93.94% | 66.86% | 82.35% | 88.19% | 75.80% | 81.43% |
| Chain | Y | N | 93.13% | 72.10% | 83.01% | 82.50% | 73.20% | 80.79% |
| Tree | Y | N | 94.91% | 71.36% | 81.74% | 77.50% | 73.60% | 79.82% |
| Complete Graph | Y | N | 94.64% | 68.60% | 82.60% | 83.75% | 74.20% | 80.76% |
| LLM-Debate | Y | N | 94.66% | 64.68% | 81.04% | 84.38% | 73.60% | 79.67% |
| GPTSwarm | Y | N | 94.66% | 68.75% | 84.25% | 86.28% | 75.40% | 81.87% |
| Agentprune | Y | N | 93.88% | 73.54% | 83.10% | 82.55% | 75.80% | 81.77% |
| AFlow | Y | N | 94.91% | 73.00% | 84.12% | 85.69% | 76.00% | 82.74% |
| LLM Pool | | | | | | | | |
| PromptLLM | N | Y | 93.92% | 73.03% | 78.43% | 86.33% | 73.60% | 81.06% |
| RouteLLM | N | Y | 93.42% | 71.29% | 81.04% | 83.85% | 72.60% | 80.44% |
| RouterDC | N | Y | 93.68% | 73.46% | 82.01% | 87.75% | 75.20% | 82.42% |
| MasRouter | Y | Y | 95.45% | 75.42% | 84.25% | 90.62% | 84.00% | 85.93% |
| **AMRO (Ours)** | Y | Y | **96.10%** | **77.12%** | **85.00%** | **91.02%** | **85.24%** | **86.90%** |

## 4.2 MAIN RESULTS

We compare AMRO with 17 baseline methods across five widely used benchmarks, as summarized in Table 1. Several consistent patterns emerge. First, without routing design, multi-agent methods generally surpass single-agent ones, as distributed collaboration enables task decomposition and reduces reliance on global historical information, thereby improving robustness. Within this setting, AMRO shows clear superiority: compared with the strongest multi-agent baseline without routing, AFlow (82.74% average accuracy), AMRO achieves 86.90%, an improvement of 4.16 absolute points. This highlights the value of dynamic routing and validates the role of pheromone-guided path selection.

Second, the results also show that AMRO outperforms routing-enabled baselines. It achieves the highest average pass@1 (86.90%), exceeding MASRouter (85.93%) and RouterDC (82.42%). The advantage is especially clear on the MATH dataset, where AMRO reaches 77.12%, surpassing MAS-Router by 1.7%. These findings indicate that the pheromone-driven probabilistic mechanism both adapts node selection to current states and reinforces high-quality paths through positive feedback, providing more competitive adaptability than methods relying on fixed allocation strategies or simple heuristic rules.

Figure 3 shows the inference time and accuracy of AMRO on GSM8K with 20–1000 parallel processes. As processes increase from 20 to 200, inference time decreases from 64.16 to 23.05 minutes in a near-linear trend, while accuracy stays stable between 94.4%–95.0%. Beyond 200 processes, accuracy even slightly improves, indicating that the load-decay– and pheromone-based path selection strategy effectively prevents overload and ensures robust performance under high concurrency.

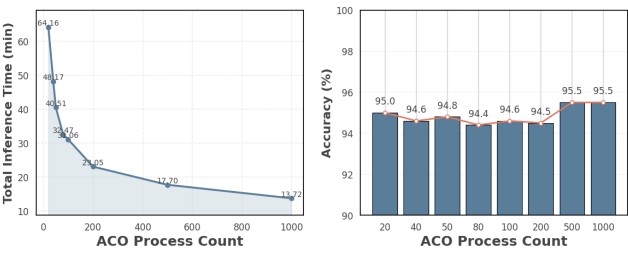

Figure 3: Impact of Parallel Process Scaling on the Inference Time and Accuracy of AMRO on GSM8K.

Table 2: Ablation results of - w/ and w/o Routing on five benchmarks.

| Model | Mul. | Rout. | GSM8K | MATH | MMLU | HumanEval | MBPP | Avg. |
|---|---|---|---|---|---|---|---|---|
| gpt-4o-mini | N | N | 93.68% | 78.43% | 81.00% | 86.60% | 79.00% | 83.72% |
| claude-3.5-haiku | N | N | 93.17% | 66.09% | 77.81% | 72.05% | 72.20% | 76.62% |
| gemini-1.5-flash | N | N | 92.67% | 74.39% | 80.04% | 80.75% | 73.00% | 80.17% |
| llama-3.1-70b | N | N | 92.58% | 74.00% | 79.08% | 80.75% | 75.80% | 80.44% |
| w/o Routing | Y | N | 92.88% | 76.25% | 78.31% | 79.20% | 77.02% | 80.73% |
| **AMRO (Ours)** | Y | Y | **96.10%** | **77.12%** | **85.00%** | **91.02%** | **85.24%** | **86.90%** |

To evaluate the transferability and general benefits of AMRO, we integrate it in a non-intrusive manner into four representative multi-agent and collaborative reasoning frameworks: MAD, Mac-Net, GPTSwarm, and HEnRY. We conduct evaluations on three benchmarks (MMLU, HumanEval, and GSM8K) under two widely used lightweight LLM configurations: gpt (gpt-4o-mini) and gemini (gemini-1.5-flash). Under identical computational and billing settings, we replace the native routing or selection strategies of each framework with AMRO, resulting in a "Framework + AMRO" configuration. At each decision point, AMRO probabilistically selects the next agent and underlying LLM according to its pheromone-based routing policy, thereby enabling dynamic routing across models and agents.

Table 2 presents the ablation results of AMRO with and without the routing mechanism. The single-agent baselines (GPT-4o-mini, Claude-3.5-haiku, Gemini-1.5-flash, and LLaMA-3.1-70B) achieve average accuracies between 76.20% and 80.54%, with GPT-4o-mini performing the best at 80.25However, when extending to a multi-agent setting without routing ("w/o Routing"), the performance drops to 77.48%, indicating that naive multi-agent collaboration does not yield consistent benefits and may even introduce coordination overhead that harms effectiveness. In contrast, incorporating our pheromone-guided routing mechanism significantly improves performance: AMRO achieves an average accuracy of 86.90%, surpassing the best single-agent baseline (GPT-4o-mini) by 6.65% and the multi-agent setting without routing by 9.42%. This improvement is consistent across GSM8K, MATH, MMLU, HumanEval, and MBPP, demonstrating the robustness and generality of our approach. These results indicate that the performance improvement of AMRO is not solely attributable to the multi-agent structure, but also critically guided by the pheromone-based routing mechanism.

Table 3: Comparison of performance and cost before and after integrating with our method. gpt and gemini are abbreviations for gpt-4o-mini and gemini-1.5-flash, respectively.

| Data | Meth | LLM | Perf | Cost |
|---|---|---|---|---|
| MMLU | MAD | gpt | 81.50 | $25.56 |
| | | gemini | 80.94 | $27.02 |
| | +MasRouter | | 82.20 | $19.39 |
| | +AMRO | | **83.70** | **$19.01** |
| HumEval | MAD | gpt | 86.05 | $1.248 |
| | | gemini | 82.95 | $1.526 |
| | +MasRouter | | 87.60 | $1.096 |
| | +AMRO | | **88.12** | **$1.032** |
| GSM8K | MAD | gpt | 94.60 | $5.664 |
| | | gemini | 94.40 | $5.492 |
| | +MasRouter | | 94.91 | $4.702 |
| | +AMRO | | **95.32** | **$4.679** |
| MMLU | MacNet | gpt | 82.98 | $7.812 |
| | | gemini | 81.74 | $8.482 |
| | +MasRouter | | 83.40 | $5.892 |
| | +AMRO | | **83.50** | **$5.500** |
| HumEval | MacNet | gpt | 86.82 | $0.488 |
| | | gemini | 88.72 | $0.568 |
| | +MasRouter | | 88.37 | $0.404 |
| | +AMRO | | **89.00** | **$0.377** |
| GSM8K | MacNet | gpt | 94.69 | $2.142 |
| | | gemini | 94.31 | $2.016 |
| | +MasRouter | | 94.89 | $1.774 |
| | +AMRO | | **95.00** | **$1.658** |

As shown in Table 3, both AMRO and MasRouter improve performance over the baseline frameworks, but AMRO achieves consistently larger gains with lower costs. In MAD, AMRO outperforms MasRouter on MMLU, HumanEval, and GSM8K, improving accuracy by an additional 1.50%, 0.52%, and 0.41%, while further reducing costs by $0.38, $0.064, and $0.023. Similarly, in MacNet, AMRO surpasses MasRouter with accuracy improvements of 0.10%, 0.63%, and 0.11%, alongside additional cost reductions of $0.392, $0.027, and $0.116. These consistent advantages indicate that

AMRO not only inherits the efficiency benefits of multi-agent routing but also optimizes accuracy–cost trade-offs more effectively than MasRouter, demonstrating superior robustness and scalability.

### 4.3 SENSITIVITY ANALYSIS OF AMRO HYPERPARAMETERS

To systematically evaluate the stability of AMRO under varying configurations, we conduct a sensitivity analysis on five key hyperparameters: pheromone weight $\alpha$ (determining the influence of pheromone concentration on path selection), heuristic weight $\beta$ (controlling the sensitivity of routing to response latency), load decay weight $\delta$ (modulating the contribution of node load to path scoring), node capacity weight $\lambda$ (reflecting the importance of model capacity in routing decisions), and pheromone evaporation rate $\rho$ (governing the forgetting rate of historical pheromones and convergence stability). These parameters correspond to distinct regulatory dimensions in AMRO's probabilistic routing mechanism, and their variations may significantly impact the system's exploration–exploitation balance, load distribution, and final convergence quality.

Table 4 presents the average accuracy (*Avg Pass@1*) and the relative change in inference cost compared to the baseline across different parameter settings. The results demonstrate that AMRO exhibits strong robustness to hyperparameter variations: within reasonable ranges, accuracy fluctuations typically remain within $\pm 2\%$, and inference cost variations stay manageable. Specifically, adjustments to $\alpha$ and $\beta$ primarily affect the trade-off between performance and latency; moderate increases in these parameters maintain stable accuracy while reducing costs, though excessively high values may induce convergence oscillations. The parameters $\delta$ and $\lambda$ mainly influence cost allocation, with limited impact on accuracy. In contrast, $\rho$ proves to be the most sensitive parameter: excessively low values lead to pheromone over-accumulation and local optima entrapment, while overly high values cause rapid forgetting of historical information, both resulting in performance instability.

Overall, the analysis indicates that the default configuration ($\alpha = 1, \beta = 1, \delta = 0.5, \lambda = 0.5, \rho = 0.1$) achieves an optimal balance between performance and cost. These findings validate AMRO's robustness and transferability, as the system maintains stable performance even without meticulous hyperparameter tuning. This not only provides flexible tuning options for practical applications but also establishes a reliable theoretical foundation for deploying AMRO across diverse scenarios.

Table 4: Sensitivity of $\alpha, \beta, \delta, \lambda, \rho$ on *Avg Pass@1* and *Cost* (relative to baseline).

| Param | Value | Avg Pass@1 | Cost |
|---|---|---|---|
| $\alpha$ | 0.5 | ↓0–1.0% | ↑3–8% |
| | 0.75 | ±0.5% | ↑1–3% |
| | 1 | baseline | baseline |
| | 1.25 | ↑0–0.8% | ↓2–5% |
| | 1.5 | ↓0–1.2% | ↓4–8% |
| | 2 | ↓1–2% | ↓6–10% |
| $\beta$ | 0.5 | ↓0.8–1.5% | ↑3–6% |
| | 0.75 | ±0.3% | ↑1–3% |
| | 1 | baseline | baseline |
| | 1.25 | ↑0.3–0.8% | ↓1–3% |
| | 1.5 | ↓0.5–1.2% | ↓3–6% |
| | 2 | ↓1–2% | ↓5–9% |
| $\delta$ | 0 | ↓1–2% | ↑5–8% |
| | 0.25 | ↓0.5–1% | ↑3–6% |
| | 0.5 | baseline | baseline |
| | 0.75 | ↓0.5–1% | ↑1–3% |
| | 1 | ↑0.3–0.8% | ↓1–3% |
| | 1.5 | ↓0.5–1.2% | ↓3–6% |
| $\lambda$ | 0 | ↓1–2% | ↑5–8% |
| | 0.25 | ↓0.5–1% | ↑2–4% |
| | 0.5 | baseline | baseline |
| | 0.75 | ↑0.5–1% | ↓1–3% |
| | 1 | ↑0.8–1.5% | ↓3–5% |
| | 1.5 | ↓0.5–1.5% | ↓4–6% |
| $\rho$ | 0.05 | ↓0–0.6% | ↓2–4% |
| | 0.1 | baseline | baseline |
| | 0.2 | ↑0.3–0.9% | ↑4–9% |
| | 0.3 | ↓0.3–1.2% | ↑8–15% |

### 5 CONCLUSION

LLM-driven MAS face significant challenges in dynamic and resource-constrained environments, particularly in task routing transparency and the inefficiency of static allocation strategies. To address these issues, we propose AMRO, which guides agent node selection probabilistically through a pheromone update mechanism and incorporates an adaptive pheromone decay strategy to enable dynamic responses to environmental changes. Its path selection strategy, based on the load decay factor and pheromone mechanism, effectively handles high-concurrency scenarios by preventing node overload and ensuring balanced resource allocation, demonstrating strong scalability and robustness in complex network environments. Extensive performance evaluations and visualization experiments demonstrate AMRO's strong effectiveness and its interpretability in the routing planning process. Overall, AMRO offers a scalable solution for transparent, efficient task routing in complex intelligent systems via bio-inspired multi-agent coordination.

## REPRODUCIBILITY STATEMENT

To ensure the reproducibility of our work, we provide comprehensive details and resources in the main paper and supplementary materials:

**Models and Algorithms:** The proposed AMRO algorithm and its core ant colony optimization mechanism are formally defined with mathematical formulations in Section B, and pseudocode implementations are provided in Appendix A.3. In addition, the algorithm's parameters, notations, and update rules are systematically specified and explained in Appendices A.2

**Theory and Interpretability:** The path fitness function, pheromone update strategy, and related assumptions are explicitly described in Section B. Furthermore, Appendix A.5 presents visual explanations—such as heatmaps of pheromone dynamics and analyses of training processes—to facilitate validation and interpretation of the model's reasoning behavior.

**Experimental Setup:** Section 4 details the experimental configuration, including the publicly available datasets (GSM8K, MATH, MMLU, HumanEval, MBPP) and evaluation metrics, with explicit definitions such as the computation of pass@K.

**Sensitivity and Ablation Studies:** Section 4.3 provides sensitivity analyses of key hyperparameters, with quantitative results reported in tables to demonstrate their impact on performance and computational cost.

**Training and Supplementary Materials:** Appendix A.6 presents intermediate training dynamics across multiple epochs, including agent initialization, path selection, pheromone updates, and convergence trends. These materials offer visualizations and detailed examples to support faithful replication.

Finally, if the paper is accepted, we will release the complete source code to further facilitate replication and extension of our work by the research community.

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

# A APPENDIX

## A.1 DETAILED RELATED WORK

### A.1.1 LLM-BASED MULTI-AGENT SYSTEM ROUTING

MAS based on LLMs tackle complex tasks through collaboration and interaction among multiple agents (Dorri et al., 2018; Kasneci et al., 2023; Zheng et al., 2025). Each agent, equipped with robust language comprehension capabilities, communicates and shares information in natural language to jointly achieve objectives. This architecture enables division of labor: agents specialize in subtasks, leveraging LLMs' reasoning and comprehension abilities to interpret requirements and coordinate actions. Overall LLMs provide critical language understanding and collaboration capabilities that enhance efficiency and flexibility. Compared to single-agent systems, LLM-based multi-agent systems are lighter, more flexible, and easily scalable (Balaji & Srinivasan, 2010). Each agent focuses solely on its assigned tasks, reducing the need to retain extensive historical context. Through parallel collaboration, the system can dynamically incorporate new agents to accommodate task expansion.

In complex tasks, LLM-based MAS path planning strategies can be categorized into non-learnable (Chen et al.; Liang et al., 2023) and learnable (Lu et al., 2023; Chen et al., 2024; Ding et al., 2024; ?; Yue et al., 2025) approaches.

AGENTVERSE (Chen et al.) is a multi-agent framework that simulates human-like group collaboration for problem-solving, with the ability to dynamically adjust the agent composition based on task progression. It achieves efficient problem-solving through four stages: expert recruitment, collaborative decision-making, action execution, and outcome evaluation. MAD (Liang et al., 2023) identifies a "rigid thinking" problem in LLMs during self-reflection. To address this issue, MAD enhances reflection and reasoning accuracy by encouraging diverse thinking through the design of agent interactions and an adjudication mechanism.

In complex task environments, using non-learnable path planning strategies may not only reduce the model's adaptability to diverse tasks, but also limit its flexibility and generalization capability in dynamic scenarios. Learnable path planning strategies have become the mainstream in current research. ZOOTER (Lu et al., 2023) introduces a reward-guided routing approach that learns a routing function by extracting supervision signals from training query rewards, enabling the assignment of each query to the most specialized LLM. RouterDC (Chen et al., 2024) trains a query-based router using contrastive loss across samples and LLMs. Hybrid-LLM (Ding et al., 2024) improves reasoning efficiency by combining strengths of different LLMs through hybrid routing. RouteLLM (?) improves reasoning efficiency by combining strengths of different LLMs through hybrid routing, while MasRouter (Yue et al., 2025) handles complex routing with a three-stage cascade for collaboration mode selection, role assignment, and final routing.

While existing methods have demonstrated promising performance, practical deployment scenarios demand even greater training efficiency and accuracy. Furthermore, the inherent black-box nature of LLMs (Ajwani et al., 2024) poses significant challenges to model transparency and interpretability. To tackle these limitations, we propose a multi-agent routing framework inspired by ant colony optimization. This approach not only enhances performance and scalability through parallel agent collaboration but also improves decision traceability by providing a more interpretable routing mechanism, making it better suited for real-world, mission-critical applications.

### A.1.2 HEURISTIC PATH OPTIMIZATION

Heuristic Path Optimization (Sivanandam et al., 2008; Rutenbar, 1989) is a rule-based approach designed to efficiently find near-optimal paths in complex spaces. Unlike exhaustive search methods that aim to evaluate all possible paths, heuristic strategies rely on informed rules or prior knowledge—such as distance estimations, gradient information, or problem-specific heuristics—to guide

---

**Algorithm 1** AMRO for MAS

---

Graph $G = (V, E)$; initial pheromone $\tau_0$; parameters $\alpha, \beta, \delta, \lambda, \mu$;
evaporation rate $\rho$; load decay rate $\beta$  Optimal Routing Allocation $P_{\text{best}}$
**Initialize:**   Set $\tau_{l,i,j} \leftarrow \tau_0$ for each edge $(v_{l,i}, v_{l+1,j}) \in E$
**Parallel and Asynchronous Agent Processing:**

**while** new agent request $a_k$ dynamically arrives **do** In parallel for all active agents
**Edge Selection:**   Start at source node $v_{1,s}$

    **for** $l = 1 \ N-1$ **do** Choose next node $v_{l+1,j}$ with probability $p_{l,i,j}$ based on pheromone, heuristic,
load, and capability   Update current node to $v_{l+1,j}$
**Fitness Evaluation:**   Compute path fitness $f(P_k)$ based on delay, success, and load
**Pheromone Update:**
edge $(v_{l,i}, v_{l+1,j}) \in P_k$ Compute $\Delta\tau_{l,i,j} = \frac{Q}{f(P_k)+\epsilon}$   Update $\tau_{l,i,j} \leftarrow (1-\rho)\cdot\tau_{l,i,j} + \sum_{k=1}^{m} \Delta\tau_{l,i,j}$
**Load Management:**
node $(v_{l,i}, v_{l+1,j})$ Compute load factor $\alpha_{l,i,j}$   Update decay factor $\gamma_{l,i,j} \leftarrow e^{-\beta\alpha_{l,i,j}}$

**return** $P_{\text{best}}$ with fitness $f(P_{\text{best}})$

---

the search process toward promising regions of the solution space. It is widely used in tasks such as robotic navigation (Tan et al., 2021), graph search (Pohl, 1970) and other tasks (Tahir et al., 2019).

ACO algorithm draws inspiration from the foraging behavior of real ant colonies (Blum, 2005; Dorigo & Socha, 2018).. In nature, ants communicate indirectly by depositing pheromones along their paths. Trails with higher pheromone concentrations are more likely to be followed, as they often represent more efficient routes. This process creates a positive feedback loop, gradually guiding the colony to converge on the shortest path between the nest and a food source. AddACO (Scianna, 2024) guides ant path selection through a linear convex combination of multiple migratory cues, such as pheromones and distance. DYACO (Liang et al., 2024) improves path search efficiency and global optimization capability by introducing guided direction, Gaussian distribution functions, and a corner-turning heuristic, while also reducing redundant path selection. PACO (Si & Bao, 2024) is applied to robot path planning in a grid network. It overcomes the local optimum limitation of traditional ACO by introducing enhanced pheromone update schemes and hybrid strategies, while significantly accelerating path planning through parallel computing.

Despite the success of ant colony algorithms in other domains, their application to LLM routing planning remains underexplored. This paper presents the first integration of ant colony optimization into LLM routing. By redesigning pheromone representations and state transition rules, we develop a distributed routing framework tailored to high-dimensional semantic spaces, offering a biologically inspired solution to traffic allocation and path optimization challenges in LLM inference.

## A.2  NOTATION

Table A.2 provides a comprehensive overview of the notations and character representations used throughout this study.

To facilitate clarity and consistency in the mathematical formulation of our proposed method, the table categorizes these representations into five distinct sections:(1) Graph Structure; (2) Path Selection and Fitness; (3) Cost, Load, and Capability; (4)Pheromone Mechanism; (5) Algorithm Parameters.

Table 5: Categorized Notations in the AMRO Method

### Graph Structure

| | |
|---|---|
| $G = (V, E)$ | Layered directed graph representing the multi-agent system |
| $v_{l,i}$ | The $i$-th node in the $l$-th layer (an agent node) |
| $(v_{l,i}, v_{l+1,j})$ | Directed edge from $v_{l,i}$ to $v_{l+1,j}$ |
| $P_k$ | Path constructed by agent $k$ |

### Path Selection and Fitness

| | |
|---|---|
| $p_{l,i,j}$ | Probability of choosing $v_{l+1,j}$ from $v_{l,i}$ |
| $f(P_k)$ | Fitness of path $P_k$ considering delay, success, and load |
| $w_1, w_2, w_3$ | Weights for token cost, failure, and load in $f(P_k)$ |

### Cost, Load, and Capability

| | |
|---|---|
| $l_{l,i,j}$ | Normalized token cost on edge $(v_{l,i}, v_{l+1,j})$ |
| $\alpha_{l,i,j}$ | Load ratio: tasks assigned / capacity of node $v_{l+1,j}$ |
| $\gamma_{l,i,j}$ | Load decay factor: $e^{-\beta \alpha_{l,i,j}}$ |
| $s_{l,i,j}$ | Normalized capability score of node $v_{l+1,j}$ |
| $\eta_{l,i,j}$ | Heuristic factor (inverse of response time of $v_{l+1,j}$) |

### Pheromone Mechanism

| | |
|---|---|
| $\tau_{l,i,j}(t)$ | Pheromone value on edge $(v_{l,i}, v_{l+1,j})$ at time $t$ |
| $\Delta\tau_{l,i,j}^k$ | Pheromone increment on edge by agent $k$ |
| $\rho$ | Global pheromone evaporation rate |
| $\xi$ | Local pheromone update coefficient |
| $Q$ | Scaling factor for pheromone reinforcement |

### Algorithm Parameters

| | |
|---|---|
| $m$ | Number of agents (ants) in one iteration |
| $T$ | Number of total iterations |
| $\epsilon$ | Small constant to avoid division by zero |

---

**Algorithm 2** Ant Colony Optimization (ACO)

---

Graph $G = (V, E)$, ants $m$, iterations $T$, evaporation rate $\rho$, initial pheromone $\tau_0$, parameters $\alpha$, $\beta$, constant $Q$ Best solution found

Initialize $\tau(i,j) \leftarrow \tau_0$, $\eta(i,j)$ for all $(i,j) \in E$ Initialize best solution and cost

**for** $t \leftarrow 1$ $T$ **do**

    **for** $k \leftarrow 1$ $m$ **do** Place ant $k$ on a random node

        **while** solution not complete **do** Select next node $j$ from $N_i$ with probability:

$$P(i,j) = \frac{[\tau(i,j)]^\alpha [\eta(i,j)]^\beta}{\sum_{l \in N_i} [\tau(i,l)]^\alpha [\eta(i,l)]^\beta}$$

Append $j$ to solution Compute cost $C_k$ Update best solution if improved

        **for** each $(i,j) \in E$ **do** $\tau(i,j) \leftarrow (1 - \rho) \cdot \tau(i,j)$

            **for** $k \leftarrow 1$ $m$ **do**

                **for** each $(i,j)$ in ant $k$'s solution **do** $\tau(i,j) \leftarrow \tau(i,j) + \frac{Q}{C_k}$     **return** best solution

---

A.3   PSEUDOCODE

---

**Algorithm 3** AMRO for MAS

---

Layered graph $G = (V, E)$, number of agents $m$, source-destination pairs, parameters $\alpha, \beta, \delta, \lambda, \mu, \rho, Q, \xi, \tau_0, w_1, w_2, w_3$ Optimized routing paths for each agent

**Initialization:**

edge $(v_{l,i}, v_{l+1,j}) \in E$ Set pheromone $\tau_{l,i,j}(0) \leftarrow \tau_0$;

**for** agent $k = 1$ to $m$ **do** Initialize path $P_k \leftarrow [v_{1,s}]$; set current node $v_{l,i} \leftarrow v_{1,s}$;

    **while** $v_{l,i} \neq v_{N,d}$ **do** Compute transition probabilities to next layer:

$$p_{l,i,j}^k(t) = \frac{[\tau_{l,i,j}(t)]^\alpha \cdot [\eta_{l,i,j}]^\beta \cdot [\gamma_{l,i,j}]^\delta \cdot [s_{l,i,j}]^\lambda \cdot [l_{l,i,j}]^{-\mu}}{\sum\limits_{v_{l+1,k} \in N_{l+1}} [\tau_{l,i,k}(t)]^\alpha \cdot [\eta_{l,i,k}]^\beta \cdot [\gamma_{l,i,k}]^\delta \cdot [s_{l,i,k}]^\lambda \cdot [l_{l,i,k}]^{-\mu}}$$

Select next node $v_{l+1,j} \sim p_{l,i,j}^k(t)$;

Append $v_{l+1,j}$ to $P_k$; set $v_{l,i} \leftarrow v_{l+1,j}$;

Update pheromone locally:

$$\tau_{l,i,j}(t) \leftarrow (1 - \xi) \cdot \tau_{l,i,j}(t) + \xi \cdot \tau_0$$

Compute fitness of path $P_k$:

$$f(P_k) = w_1 \cdot \sum_{(l,i,j) \in P_k} l_{l,i,j} + w_2 \cdot \left(1 - \prod_{(l,i,j) \in P_k} s_{l,i,j}\right) + w_3 \cdot \frac{1}{|P_k|} \sum_{(l,i,j) \in P_k} \gamma_{l,i,j}$$

edge $(v_{l,i}, v_{l+1,j}) \in E$ Compute pheromone increment:

$$\Delta\tau_{l,i,j}^k = \begin{cases} \frac{Q}{f(P_k) + \epsilon}, & \text{if } (v_{l,i}, v_{l+1,j}) \in P_k \\ 0, & \text{otherwise} \end{cases}$$

Update pheromone globally:

$$\tau_{l,i,j}(t+1) \leftarrow (1 - \rho) \cdot \tau_{l,i,j}(t) + \sum_{k=1}^{m} \Delta\tau_{l,i,j}^k$$

**(Optional) Adaptive evaporation:**

$$\rho_t \leftarrow \rho \cdot \left(1 + \frac{\Delta\bar{\gamma}}{\bar{\gamma}}\right)$$

**return** Paths $P_k$ with the lowest fitness or highest accumulated pheromone

---

Algorithm 2 describes the ACO algorithm, which simulates the collective foraging behavior of ants to solve combinatorial optimization problems over a graph. Initially, pheromone levels on all edges are uniformly set, and each ant incrementally constructs a solution by probabilistically selecting the next node based on pheromone intensity and heuristic desirability. After all ants complete their paths, pheromone evaporation is applied to prevent convergence to suboptimal solutions, followed by pheromone reinforcement based on the quality of each ant's solution. This process iterates over multiple rounds, progressively guiding the search toward high-quality solutions. The best solution found during the process is returned as the final output.

AMRO is specifically designed for hierarchical graph structures, targeting task routing and path planning problems in multi-agent systems. At its core, AMRO probabilistically guides agents to discover optimal paths through a layered graph using pheromone-based decision-making, while integrating multiple factors such as node load, capacity, and heuristic desirability. This results in an efficient and interpretable solution for cooperative multi-agent path optimization.

The algorithm begins by initializing the pheromone value $\tau_0$ for every edge in the graph. During each iteration, every agent starts from a designated source node and constructs a path toward a destination

node. At each step, the agent selects the next-hop node from the adjacent layer by computing a transition probability $p_{l,i,j}^k(t)$ that depends on the pheromone strength $\tau$, heuristic factor $\eta$, node capacity $\gamma$, path feasibility $s$, and path load $l$. Local pheromone updates are performed immediately after each move to reflect recent exploration behavior.

Once a complete path is constructed for each agent, a fitness score $f(P_k)$ is computed, which incorporates weighted metrics such as total path length, availability penalties, and average node capacity. After all agents complete their path construction, global pheromone updates are applied across the graph. Edges that appear in successful paths are reinforced proportionally to the pheromone increment $\Delta\tau$ and the corresponding fitness value $f(P_k)$, while non-utilized edges undergo pheromone evaporation. To improve adaptability to dynamic environments, AMRO also supports an adaptive pheromone evaporation mechanism, which adjusts the evaporation rate $\rho$ in response to real-time system state changes.

Finally, the set of paths $P_k$ with the highest pheromone accumulation or optimal fitness values are selected as the final output. By jointly optimizing path quality, system load balancing, and resource allocation, AMRO demonstrates robust convergence and scalability, making it well-suited for high-concurrency, multi-agent task scenarios.

### A.4 MORE EXPERIMENTAL RESULTS

#### A.4.1 BASELINE

**Single-Agent Methods**

CoT (Chain of Thought) guides the model to decompose logic step-by-step and generate coherent reasoning processes when tackling complex tasks by embedding intermediate reasoning steps into input problems, thereby enhancing the ability to solve intricate problems. ComplexCoT uses high-complexity chain-of-thought examples during prompting and selects the majority answer from multiple generated reasoning chains during decoding, significantly improving model accuracy for multi-step reasoning tasks. SC(COT) and SC(ComplexCOT) adopt a decoding strategy that combines diverse reasoning path sampling with self-consistent answer selection, generating multiple sets of reasoning chains and choosing the majority-consistent answer.

**Multi-Agent Methods** LLM-debate organizes multiple large language models to simulate human multi-annotator collaborative evaluation processes, forming an autonomous debating jury to collaboratively assess the quality of generated responses. GPTSwarm models LLM agents and their interaction relationships through computational graphs, integrating a node-level prompt optimizer with an automatic graph optimizer that adjusts graph connections. Agent-prune identifies communication redundancy and performs one-time pruning on spatio-temporal message-passing graphs, maintaining high model performance while significantly reducing token overhead and economic costs. Within the three structures of the multi-agent collaboration network MACNET—Chain, Tree, and Complete Graph—Chain arranges agents in a linear sequence with sequential interactions; Tree allows agents to interact along different branches; and Complete Graph fully connects each node to enable arbitrary interaction dependencies, facilitating dense information propagation and diverse interactions. AFlow employs a Monte Carlo tree search mechanism to automate the optimization of coded workflows, achieving efficient generation and refinement of complex task workflows through iterative code modification, tree-structured experience accumulation, and execution feedback.

**Learnable Single-Agent Routing Methods** FrugalGPT proposes a lightweight and efficient LLM cascading framework that dynamically selects the optimal LLM combination for different queries through learning. PromptLLM incorporates query content, candidate models, and target requirements into prompt text, which is then input to an external large language model to screen for the most suitable candidate model. RouteLLM utilizes a dynamic mechanism to select between strong and weak LLMs, combined with a training framework based on human preference data and data augmentation, to optimize the balance between cost and response quality during inference. RouterDC presents a dual-contrastive learning-based approach that leverages encoders, LLM embeddings, and two contrastive learning losses to achieve effective planning across multiple LLMs.

| Model | Epoch 1 COT | TOT | GOT | AOT | Epoch 2 COT | TOT | GOT | AOT | Epoch 3 COT | TOT | GOT | AOT | Epoch 4 COT | TOT | GOT | AOT |
|---|---|---|---|---|---|---|---|---|---|---|---|---|---|---|---|---|
| claude-3.5-haiku | 0.37 | 0.28 | 0.29 | 0.22 | 0.49 | 0.39 | 0.35 | 0.32 | 0.72 | 0.58 | 0.51 | 0.43 | 0.85 | 0.73 | 0.64 | 0.58 |
| gemini-1.5-flash | 0.95 | 1.05 | 1.10 | 0.92 | 1.02 | 1.15 | 1.23 | 1.05 | 1.20 | 1.33 | 1.38 | 1.14 | 1.35 | 1.48 | 1.55 | 1.37 |
| llama-3.1-70b | 1.30 | 1.10 | 1.10 | 1.08 | 1.71 | 1.45 | 1.42 | 1.30 | 2.01 | 1.75 | 1.60 | 1.42 | 2.55 | 2.23 | 2.08 | 1.86 |
| gpt-4o-mini | 1.72 | 1.28 | 1.41 | 1.34 | 2.13 | 1.92 | 2.12 | 1.98 | 3.02 | 2.45 | 2.78 | 2.23 | 4.11 | 3.35 | 3.70 | 3.15 |
| llama-3.1-70b | 0.56 | 0.58 | 0.49 | 0.45 | 0.69 | 0.70 | 0.62 | 0.58 | 0.75 | 0.74 | 0.69 | 0.63 | 0.81 | 0.80 | 0.74 | 0.70 |
| gpt-4o-mini | 0.94 | 0.77 | 0.88 | 0.74 | 1.10 | 0.84 | 0.95 | 0.83 | 1.53 | 1.22 | 1.31 | 1.22 | 1.72 | 1.56 | 1.66 | 1.58 |
| claude-3.5-haiku | 1.05 | 0.92 | 0.88 | 0.78 | 1.11 | 0.98 | 0.94 | 0.88 | 1.29 | 1.15 | 1.02 | 0.96 | 1.41 | 1.28 | 1.15 | 1.04 |
| gemini-1.5-flash | 1.13 | 1.22 | 1.03 | 1.11 | 1.27 | 1.42 | 1.14 | 1.22 | 1.45 | 1.57 | 1.28 | 1.37 | 1.71 | 1.83 | 1.56 | 1.50 |
| claude-3.5-haiku | 0.29 | 0.22 | 0.17 | 0.23 | 0.39 | 0.33 | 0.22 | 0.28 | 0.55 | 0.49 | 0.37 | 0.43 | 0.66 | 0.59 | 0.49 | 0.52 |
| gpt-4o-mini | 0.82 | 0.58 | 0.64 | 0.47 | 1.03 | 0.78 | 0.85 | 0.63 | 1.22 | 0.95 | 1.05 | 0.80 | 1.82 | 1.45 | 1.62 | 1.35 |
| llama-3.1-70b | 1.22 | 1.05 | 0.93 | 1.03 | 1.35 | 1.18 | 1.08 | 1.10 | 1.45 | 1.30 | 1.18 | 1.15 | 1.60 | 1.47 | 1.36 | 1.32 |
| gemini-1.5-flash | 1.35 | 1.45 | 1.10 | 1.15 | 1.45 | 1.58 | 1.22 | 1.25 | 1.56 | 1.72 | 1.45 | 1.35 | 1.89 | 2.02 | 1.75 | 1.55 |

Figure 4: Heatmaps of pheromone-guided routing across four epochs, where color intensity denotes model–method performance and AMRO progressively shifts from broad exploration to reinforced high-performing paths while suppressing weaker ones.

## A.5 INTERPRETABILITY OF AMRO

To evaluate AMRO's interpretability, we visualize its pheromone update dynamics across four training epochs using heatmaps. As shown in Figure 4, the heatmaps illustrate the evolution of scores for combinations of models and methods (e.g., CoT, ToT, GoT, AoT), with color intensity reflecting the preference strength for selection. The scores evolve progressively, depicting AMRO's transition from exploration to convergence.

In the initial stage (Epoch 1), scores are relatively uniform, indicating robust exploration and avoidance of premature convergence. During intermediate stages (Epochs 2–3), high-performing combinations, such as `gpt-4o-mini` with ToT and GoT, show increasing scores, with deeper colors reflecting pheromone accumulation and higher selection probability. Conversely, weaker combinations, such as `claude-3.5-haiku` with AoT, exhibit declining scores and are gradually marginalized. By the convergence stage (Epoch 4), top-performing combinations dominate with peak scores, while inefficient ones are naturally eliminated.

This evolution underscores AMRO's pheromone-guided probabilistic path selection: successful paths accumulate pheromones through positive feedback, increasing their attractiveness, while inefficient paths decay due to evaporation. AMRO thus balances exploration and exploitation, initially exploring diverse combinations and later converging to an optimized and stable routing strategy.

## A.6 TRAINING EXAMPLES

This appendix presents three training epoch examples to systematically illustrate the performance evolution trend of the AMRO framework during task scheduling and how its ACO-based path selection mechanism iteratively optimizes through successive iterations. The content includes: initial agent configurations, schematic diagrams of task path selection, dynamic pheromone updates, the model performance evolution process, and final path convergence with agent weight distribution.

### A.6.1 EPOCH 1

**Initialization Phase** Before the start of the first training round, the system performs parameter initialization for all 30 available Agents. The dynamic weight and pheromone concentration of each Agent are both set to 1.00, with only static weights differing due to variations in model capabilities and role functions. Concentration The initial parameter configurations of each agent are shown in Figure 1.

---

**The initial parameter configurations of each agent**

**GPT-4o-mini:**
**Prompt types:**
Chain-of-Thought (COT), Tree-of-Thought (TOT), Root Cause Analysis (RCA), Reverse Thinking (Reverse), Analogical Reasoning (Analogy), Hypothetical Deduction (Hypothesis), and Six Thinking Hats (SixHats)

---

**Initial weight:**
Static weight of 2.71.
Dynamic weight of 1.00.
Initial pheromone concentration of 1.00.
Comprehensive weight of 2.71.

**GPT-4.1-nano:**
**Prompt types:**
COT, TOT, RCA, Reverse, Analogy, Hypothesis, and SixHats
**Initial weight:**
Static weight of 2.38.
Dynamic weight of 1.00.
Initial pheromone concentration of 1.00. omprehensive weight of 2.38.

**Deepseek-chat:**
**Prompt types:**
First Principle, COT
**Initial weight:**
Static weight of 1.83.
Dynamic weight of 1.00.
Initial pheromone concentration of 1.00.
Comprehensive weight of 1.83.

**Qwen2.5-coder-7b-instruct :**
**Prompt types:**
TOT
**Initial weight:**
Static weight of 1.38.
Dynamic weight of 1.00.
Initial pheromone concentration of 1.00.
Comprehensive weight of 1.38.

In the initial phase, the current load of all Agents is 0, with a maximum task capacity of 10, indicating that each Agent in the system has sufficient computational resources to execute tasks. Due to the highest static weight (2.71) among all models, GPT-4o-mini Agents have the highest comprehensive weight and thus a significant advantage in initial path selection. In contrast, Qwen Agents have the lowest static weight (1.38), resulting in a relatively lower selection probability before feedback mechanisms are introduced.

**Path Selection and Load Balancing Based on Comprehensive Weight** As the Epoch progresses, the system dynamically selects execution paths for tasks based on comprehensive weights. Specifically, when a new task requires allocation, the system calculates the comprehensive weights of all candidate Agents and selects them in proportion to these weights—the greater an Agent's comprehensive weight, the higher the probability of being chosen as the next node in the path. However, the system also achieves load balancing through dynamic weights to prevent high-weight Agents from monopolizing tasks.

**Agent Selection Process in Epoch 1**

**Layer1 data_collection:**
**Initial weight:**
GPT-4o-mini-Hypothesis Agent of 2.71 (**selected**)
GPT-4o-mini-TOT Agent of 2.71
...
GPT-4.1-nano-SixHats Agent of 2.38
...

Deepseek-chat-FirstPrinciple Agent of 1.83

...

Qwen2.5-coder-7b-instruct-TOT Agent of 1.38

**Layer2 Data_analysis:**
GPT-4o-mini-Hypothesis Agent of 2.71
GPT-4o-mini-TOT Agent of 2.71

...

GPT-4.1-nano-SixHats Agent of 2.38 (**selected**)

...

Deepseek-chat-FirstPrinciple Agent of 1.83

...

Qwen2.5-coder-7b-instruct-TOT Agent of 1.38

**Layer3 Answer_generation:**
GPT-4o-mini-Hypothesis Agent of 2.71
GPT-4o-mini-TOT Agent of 2.71 (**selected**)

...

GPT-4.1-nano-SixHats Agent of 2.38

...

Deepseek-chat-FirstPrinciple Agent of 1.83

...

Qwen2.5-coder-7b-instruct-TOT Agent of 1.38

**Agent Selection Process** The system first designated the GPT-4o-mini-Hypothesis agent as the first-layer processing node, given its highest comprehensive weight (2.71) and suitability for hypothesis generation. In the second layer (data_analysis), the system selected the GPT-4.1-nano-SixHats agent for multi-perspective problem analysis, leveraging the previous output and original problem. Despite sharing the same static weight (2.38) as other roles in the model, this agent was prioritized due to its idle status and high comprehensive weight. In the third layer (answer_generation), the system chose the GPT-4o-mini-TOT agent—equipped with both problem-solving and evaluation capabilities—to generate the final answer, as it had the highest comprehensive weight in this layer.

### A.6.2 Epoch 2

**Dynamic Pheromone Adjustment** After the first epoch, AMRO updates each Agent's pheromone values based on their problem-solving success status. Agents with higher success rates experience a notable increase in pheromone. For Agents with lower success rates, pheromone increases are less significant or nearly unchanged.

**Dynamic Weight Adjustment** At the start of the first Epoch, the current load of all Agents in the system is 0 (with a maximum load set to 10), and their dynamic weights are initialized to 1.00. As tasks are executed, the system assigns tasks to Agents in sequence, causing their loads to gradually increase. To reflect load changes, dynamic weights decrease proportionally to the remaining capacity.

**Agent Selection Process in Epoch 2**

**Layer1 data_collection:**
**Initial weight:**
GPT-4o-mini-Hypothesis Agent of 6.75 (**selected**)
GPT-4o-mini-TOT Agent of 6.65

...

GPT-4.1-nano-SixHats Agent of 5.63
GPT-4.1-nano-RCA Agent of 2.29 (**lower success**)

...

Deepseek-chat-FirstPrinciple Agent of 4.58

...
Qwen2.5-coder-7b-instruct-TOT Agent of 2.90

**Layer2 Data_analysis:**
GPT-4o-mini-Hypothesis Agent of 6.75
GPT-4o-mini-TOT Agent of 6.65
...
GPT-4.1-nano-SixHats Agent of 5.63 (**selected**)
GPT-4.1-nano-RCA Agent of 2.29 (**lower success**)
...
Deepseek-chat-FirstPrinciple Agent of 4.58
...
Qwen2.5-coder-7b-instruct-TOT Agent of 2.90

**Layer3 Answer_generation:**
GPT-4o-mini-Hypothesis Agent of 6.75
GPT-4o-mini-TOT Agent of 6.65 (**selected**)
...
GPT-4.1-nano-SixHats Agent of 5.63
GPT-4.1-nano-RCA Agent of 2.29 (**lower success**)
...
Deepseek-chat-FirstPrinciple Agent of 4.58
...
Qwen2.5-coder-7b-instruct-TOT Agent of 2.90

### A.6.3   EPOCH 3

After the Epoch 2 iteration, selected agents receive further weight increases, with top-performing agents having weights several times higher than some weaker ones. However, if certain agents make mistakes in the Epoch 2, leading to a drop in the weights of previously overused agents, this triggers corrective adjustments in subsequent path selections. Such adaptation is expected to improve the success rate on previously failed problems in the next round, forming a new balance. This mechanism ensures that, as training progresses, the multi-agent problem-solving system moves toward higher overall success rates and more robust decision paths.

## B   LIMITATIONS

Although AMRO demonstrates strong performance and interpretability in multi-agent LLM routing tasks, there are still some limitations. First, our work is mainly showcased within the designed hierarchical network structure. Due to time and resource constraints, we have not yet validated the method on a truly large-scale dynamic multi-agent network, nor have we demonstrated superior performance on more diverse datasets. In addition, although AMRO reduces the reliance on large-scale annotated data compared to end-to-end learning methods, it still requires careful tuning of certain parameters (such as pheromone decay rate and weight coefficients) across different domains and tasks to achieve optimal performance.

