# OpenReview forum: "Efficient and Interpretable Multi-Agent LLM Routing via Ant Colony Optimization"
_ICLR.cc/2026/Conference — ICLR 2026 Conference Withdrawn Submission_

### Official Review · Reviewer_QZPD · 2025-10-29

**Soundness:** 2
**Presentation:** 3
**Contribution:** 2
**Rating:** 6
**Confidence:** 3

**Summary:**

The paper tackles the routing bottleneck in LLM-based multi-agent systems, highlighting issues of low transparency, static allocation, and limited system awareness. The authors propose AMRO, an ant-colony-optimization–inspired routing algorithm that models agent interactions as a function-oriented graph. By leveraging pheromone-driven updates and adaptive decay, AMRO achieves real-time adaptation to dynamic environments and continuous routing optimization. The approach improves both efficiency and interpretability, outperforming baselines by 1.97% on average in pass@1 accuracy across five benchmarks, while maintaining strong scalability and robustness under high concurrency.

**Strengths:**

1. It proposes a practical, multi-objective dynamic routing mechanism. The essence of AMRO lies in a decentralized, feedback-based routing algorithm that skillfully combines historical performance (pheromones) with current states (load, latency) to dynamically balance multiple objectives, including quality, cost, speed, and load.

2. Using Ant Colony Optimization to optimize the MAS structure is more stable and incurs lower training costs compared to certain RL algorithms, and its effectiveness is demonstrated through experiments.

3. The automated construction of MAS and routing among heterogeneous LLMs represent promising research directions.

**Weaknesses:**

1. The modeling and workflow of the proposed method are not clearly introduced in the main text, leading to confusion. For example, the definition of “layer” only appears in the caption of Figure 2 and is not elaborated upon in the main text. This may easily mislead readers into thinking that the “layers” refer to the central parts of Figure 2—LLM Type, Method Type, and Role Type—which resembles the architecture of MasRouter.

2. The algorithm appears insufficient in customizing the MAS framework at the task- and query-granularity levels. AMRO yields a framework that is globally optimal for the entire training set. However, when the task types in the test set differ significantly from those in the training set (e.g., switching from code generation to mathematical reasoning), the previously optimized framework may no longer be applicable.

3. Furthermore, tasks of different difficulty levels within the same dataset might benefit from different frameworks (e.g., simpler structures for easier questions to reduce overhead and latency). AMRO seems unable to perceive such differences at the query level or to perform customized, query-specific framework adaptation.

4. The paper defines the function Ability(Model) (in Section 3.2.2, around source line 257) but never explains how the specific Ability value for each model is actually calculated.

5. The pseudocode in Algorithm 1 is disorganize: for example, lines 765, 769, and 771 mix several operations in one line. The indentation within the while loop is also inconsistent, making it difficult for readers to discern which operations are included in the loop.

**Questions:**

1. I am curious about the agent preferences (selection probability distributions) obtained by the AMRO algorithm on various datasets.
2. How many training epochs does AMRO typically require for convergence, and what is its training cost?

---

### Official Review · Reviewer_98De · 2025-10-29

**Soundness:** 3
**Presentation:** 4
**Contribution:** 2
**Rating:** 4
**Confidence:** 4

**Summary:**

This paper proposes AMRO, a method that applies Ant Colony Optimization to the routing problem in LLM-based multi-agent systems. The authors model agent interactions as a directed graph and utilize a pheromone-driven node update mechanism with adaptive decay strategies to achieve dynamic routing optimization. Experiments on five public datasets show that AMRO achieves an average improvement in performance and demonstrates reasonable efficiency and robustness under high concurrency scenarios.

**Strengths:**

1. First application of ant colony optimization to LLM-based multi-agent routing presents an interesting cross-domain perspective

2. Pheromone visualization provides some interpretability for routing decisions, improving over purely black-box LLM routing approaches

3. Load-aware routing selection mechanism shows practical value in high-concurrency testing scenarios

4. Detailed hyperparameter sensitivity analysis demonstrates the stability of the approach

**Weaknesses:**

1. ACO is a mature algorithm, and the paper primarily adapts it to LLM routing without deep algorithmic innovation. Equations 7-10 are standard ACO variants with unclear essential differences from traditional ACO

2. Compared to the strongest baseline MASRouter, average gain is only $0.97%$, with $1.7%$ improvement on MATH dataset. Such marginal improvements hardly justify the complexity of introducing ACO mechanisms

3. The system is rigidly designed as N-layer structure with n nodes per layer. This artificially fixed topology cannot well reflect the dynamism and heterogeneity

4. The paper provides no convergence proof, time complexity analysis, or approximation guarantees. Why is ACO suitable for this problem?

5. The analogy between task routing and ant foraging is oversimplified and insufficiently justified

6. Over-claims. Pheromone heatmap visualizations merely display weight distributions. Compared to attention mechanisms or causal analysis, this interpretability is superficial

7. In Table 2, w/o Routing performs abnormally poorly, suggesting potential improper baseline implementation. Cost calculation in Table 3 is unexplained, making cost-effectiveness claims difficult to verify

8. Hyperparameter tuning cost not evaluated. Table 4 shows sensitivity to parameters. So, How to select these parameters for new scenarios?

**Questions:**

1. How to initialize pheromones in cold-start scenarios without historical data?

2. Can you provide time complexity analysis? Is the additional computational overhead of ACO justified compared to simple weighted round-robin?

3. How to handle dynamic agent addition and removal?

4. How is the fitness function in Equation 8 designed? How are w1, w2, w3 determined?

5. Table 3 shows improvements when integrated into MAD and MacNet, but gains are smaller than using AMRO directly. Does this indicate incompatibility with certain multi-agent frameworks?

6. How does pheromone update frequency affect system stability?

7. Training examples in Appendix show GPT-4o-mini being frequently selected. Does this mean the system degenerates to simple model selection rather than true multi-agent collaboration?

---

### Official Review · Reviewer_Pv97 · 2025-11-01

**Soundness:** 3
**Presentation:** 3
**Contribution:** 3
**Rating:** 4
**Confidence:** 4

**Summary:**

The paper proposes AMRO, a routing framework for Large Language Model (LLM)-based Multi-Agent Systems (MAS), inspired by Ant Colony Optimization (ACO). By modeling agents and their interactions as a hierarchical graph and using pheromone-guided path selection, AMRO aims to enhance both routing efficiency and interpretability. The authors evaluate their method on five public benchmarks, demonstrating moderate performance improvements over existing routing baselines.

**Strengths:**

S1: This is the first known application of Ant Colony Optimization to the task of LLM-based MAS routing, introducing a biologically inspired mechanism into the AI routing domain.

S2: AMRO’s pheromone-guided routing introduces visualizable decision processes, addressing the black-box limitations common in LLM routing.

S3: The probabilistic routing and pheromone update processes are rigorously formalized, which aids reproducibility.


S4: Sensitivity analyses across various parameters show AMRO’s performance stability.

**Weaknesses:**

The best-case performance gains over the strongest baseline (MasRouter) are only 0.97% on average. This is modest considering the added complexity and novelty, and may not justify the overhead in practical systems.


The use of ACO in this paper is largely a standard adaptation; the methodology borrows heavily from classical ACO formulations without substantive algorithmic innovation tailored to LLM routing.


While routing optimization is valuable, the paper lacks discussion on how agent specialization or heterogeneity contributes to performance, which is critical in multi-agent systems.


The interpretability claim, while intuitively supported by pheromone trails, is not evaluated rigorously. No user studies or quantitative measures (e.g., fidelity, human trust) are provided.

**Questions:**

1. **Why Ant Colony Optimization (ACO) over other heuristics or learnable methods?**  The paper does not convincingly justify *why* ACO is more suitable than other heuristic strategies (e.g., reinforcement learning, meta-heuristics like particle swarm optimization) for MAS routing. ACO seems chosen primarily for interpretability, but other methods might offer stronger adaptability or theoretical grounding.


2. **How are agent capabilities quantified and updated during routing?** The capability score is used in edge weight calculations, but it's unclear whether this is a static or dynamic metric. How is this score computed, and does it evolve as the agent accumulates experience or data?

3. **Is the pheromone decay schedule globally optimal or empirically tuned?** The global pheromone decay rate ρ plays a critical role in balancing exploration and exploitation. Was any theoretical or automated method used to choose its value, or is it selected manually?

4. **How is scalability ensured with increasing agents and layers?** While the paper claims scalability, there is no theoretical or empirical complexity analysis. Does the pheromone matrix or routing probability become computationally expensive in larger graphs?

---

### Official Review · Reviewer_PeY1 · 2025-11-03

**Soundness:** 3
**Presentation:** 2
**Contribution:** 2
**Rating:** 4
**Confidence:** 4

**Summary:**

This paper introduces AMRO, a pheromone-guided routing strategy for LLM-based multi-agent systems. The method is inspired by ant colony optimization and models routing as a layered directed graph. Agents are connected by probabilistic paths influenced by pheromone intensity, response latency, node capability, and load. AMRO includes mechanisms for pheromone evaporation and path reinforcement to dynamically adapt to changing environments. The authors evaluate their method on five benchmark datasets and show consistent improvements in accuracy and cost-efficiency over existing baselines.

**Strengths:**

1. The method offers a clear and intuitive framework for dynamic routing in multi-agent systems.

2. Integrating pheromone-guided decisions improves interpretability and responsiveness.

3. The experimental setup is comprehensive, covering multiple datasets and baselines.

**Weaknesses:**

1. The novelty is limited. The approach transfers existing ACO techniques with minimal adaptation to the LLM domain.

2. There is no quantitative evaluation of interpretability, which is central to the paper’s motivation.

3. The method introduces many hyperparameters. Their tuning process is not well justified or automated.

4. Scalability to large agent graphs is not discussed. Real-world feasibility in latency-sensitive environments remains unclear.

**Questions:**

1. How would AMRO handle dynamic environments where agent capabilities change over time? Could the routing adapt without retraining or manual reconfiguration?

2. Can the approach scale to thousands of agents in real-time systems? What are the practical limits in terms of latency or resource usage?

---

### Note · Authors · 2025-11-13

I have read and agree with the venue's withdrawal policy on behalf of myself and my co-authors.